

# A quality control method based on physical constraints and data-driven collaborative for wind observations along high-speed railway lines

Xiong Xiong[1], Jiajun Chen[2], Yanchao Zhang[2], Xin Chen[2], Yingchao Zhang[1], Xiaoling Ye[1]

[1]Jiangsu Collaborative Innovation Center of Atmospheric Environment and Equipment Technology, Nanjing University of Information Science and Technology, Nanjing, 210044, China

[2]Information and Systems Science Institute, Nanjing University of Information Science and Technology, Nanjing, 210044, China

*Correspondence to*: Jiajun Chen (202212490634@nuist.edu.cn), Xiaoling Ye (000510@nuist.edu.cn)

**Abstract.** This study proposed a new quality control method via physical constraints and data-driven collaborative artificial intelligence (PD-BX) to reduce wind speed measurement errors caused by the complex environment along high-speed railway lines, achieving enhanced accuracy and reliability. On the one hand, based on the special structure in railway assembly, the physical constraint model of the railway electrical catenary supports and anemometers were experimentally established. The performance of the physical model in the wind field was simulated based on FLUENT software and the environmental change characteristics of the anemometer in the railway area were analyzed. On the other hand, to solve the constrained error mapping expression under different wind conditions, a data-driven model of hyperparameter optimization (BO-XGBoost) is introduced to perform error compensation on physical relationships. Through the PD-BX method, the RMSE of the railway anemometer was reduced by 2.497 from 2.790 to 0.293, achieving quality control of wind observations along the high-speed railway lines and providing reliable results for improving the accuracy of the high-speed railway early warning system.

## 1 Introduction

Since the opening of the Tokaido Shinkansen in Japan in 1964, the construction scale of high-speed railways worldwide has continued to expand, and safety issues related to high-speed railways have increasingly garnered widespread international attention. In high-speed railway operations, strong winds are one of the major meteorological disasters threatening the safety of high-speed train operations (Liu et al., 2022; Wang et al., 2021). To ensure the safety of high-speed rail operations, wind warning systems are gradually being established around the world (Liu et al., 2021). These systems rely on wind speed and direction measurements along high-speed railway lines, which impose elevated requirements.

Ultrasonic wind measurement is the optimal option for railway systems. Ultrasonic anemometers measure wind speed by utilizing the effect of wind on the propagation speed of ultrasonic waves in the air. Compared to traditional mechanical wind measurement methods, an integrated design is featured by devices. There are no moving parts during measurement, no mechanical wear, and no risk of component detachment. Additionally, a long service life, fast response, great measurement accuracy, excellent resolution, and the ability to measure high-frequency pulsations in wind speed are characteristics of these



devices. Furthermore, random error identification technology is used to ensure low measurement errors even under strong winds. This results in smoother outputs and lower maintenance costs (Pirhalla et al., 2020; Salas et al., 2022). Therefore, ultrasonic wind measurement is the preferred choice for railway wind measurement, and numerous ultrasonic wind speed devices are installed along railway lines to mitigate the adverse effects of extreme weather on train operations (Zhang et al.,

2019). During the wind measurement process, catenary support unavoidably obstructs wind speed instruments, leading to measurement errors, false alarms, missed alarms and prolonged speed restrictions. These issues may have significant impacts on operational safety and efficiency. The thorough investigation of the issue of shadow obstruction of the anemometer is therefore crucial to ensure the safety protection of high-speed trains.

Application research on the ultrasonic anemometer has encompassed several crucial areas.  Indoor environments have

been significantly improved through Computational Fluid Dynamics (CFD) and anemometers optimization, measuring air movement, energy transfer, ventilation, and pollutants within buildings (Antonini et al., 2019; Arens et al., 2020). In addition, recent research successfully reduced the deviation caused by wind interference during Unmanned Aerial Vehicles (UAV) flight by establishing a wind speed and wind direction function model, and achieved a breakthrough in the field of unmanned flight (Cho et al., 2019; Ghirardelli et al., 2023; Li et al., 2023). In atmospheric turbulence research, scholars have integrated

ultrasonic anemometers into wind profilers, capturing turbulence characteristics under different terrains, weather conditions and wind directions through improved parameter algorithms, aiding in a more comprehensive understanding and analysis of atmospheric turbulence (Mauder et al., 2020).

Although ultrasonic anemometers have been extensively studied, certain measurement errors still exist. Not only do these errors come from the structure of the ultrasonic anemometer, but they may also be caused by the external special measurement

environment (Ghahramani et al., 2019; Shan et al., 2023). Common error elimination methods include: (1) employing high-quality and high-precision sensors and components to improve measurement accuracy (Knöller et al., 2024; White et al., 2020); (2) validating the accuracy of ultrasonic anemometers through on-site calibration and comparative studies with other wind measurement devices (Lv et al., 2024; Osterwalder et al., 2020); (3) enhancing data processing algorithms, such as data filtering, machine learning and interpolation techniques, to improve measurement accuracy (Yang et al., 2024b). Among these, machine

learning models have been increasingly utilized to compensate for the shadow effect of ultrasonic anemometers, owing to their advantages of low cost, remarkable accuracy and wide applicability. It also has shown excellent quality control results for wind observations when dealing with errors caused by the external environment (Liao et al., 2020).This method involves collecting measurement data containing shadow effects, segmenting data, selecting appropriate machine learning models for training, evaluating model performance through cross-validation and performance metrics, and applying trained models to

actual measurements of ultrasonic anemometers to real-time correct errors caused by shadow effects (Thielicke et al., 2021).

Due to the bulky items of experimental objects in the high-speed railway system, wind tunnel experiments are costly and challenging to conduct. With its capability to simulate various physical fields, fluid-structure interaction, and air flow diffusion,



CFD technology has seen its application expanding continuously as the technology matures, gradually becoming one of the primary approaches for research in the railway domain (Lin et al., 2020). Compared to traditional wind tunnel tests, CFD technology is not limited by similarity criteria and wind tunnel scales. It allows for the simulation of flow fields of any size and shape, addressing some problems that traditional wind tunnel tests cannot solve, such as simulating structures and high Reynolds number flows (Golshan et al., 2020). Additionally, CFD technology enables the visualization of flow fields through powerful post-processing capabilities, facilitating an intuitive perception of flow field distribution characteristics (Calzolari and Liu, 2021). The gradual maturity of CFD technology has provided a stable and efficient method for the safety and efficiency of high-speed railway operations (Lu et al., 2024).

The current investigation into high-speed railway strong wind conditions employing CFD technology primarily focuses on the airflow around the surface of the train. It includes the aerodynamic effects of crosswinds between the train and the rails (Liu et al., 2020; Szudarek et al., 2022), the impact of wind barriers on crosswind obstruction (Deng et al., 2021; Liu et al., 2018), and the changes in airflow generated by train movement (Liang et al., 2020). These studies play an essential role in ensuring the safety of high-speed trains. However, there is a lack of literature on the application of anemometers in high-speed railways. Only the physical adjustment of the layout improves the accuracy of railway anemometers (Zhang et al., 2019). Besides, the anemometer is an indispensable device in railway operations. It is widely distributed along the railway, and almost all wind speed monitoring along the railway relies on it. The intervals between anemometers are generally only a few kilometers or even hundreds of meters, and severe wind monitoring covers the entire section. According to Chinese rail speed limit regulations, if the wind speed continuously exceeds the alarm threshold for 10 seconds, an alarm will be triggered. Moreover, when the instantaneous wind speed reaches the critical overturning wind speed, the train may derail or overturn within 1 to 2 seconds (Chen et al., 2024). Faced with practical problems such as insufficient accuracy of railway anemometers for high wind monitoring and delayed high wind warnings, it is urgent to take measures to improve the precision of wind speed measurement along high-speed railways.

This study proposes the PD-BX method, which is dedicated to improving the accuracy of railway anemometers. A physical constraint model was established with the catenary support and anemometer. Subsequently, CFD technology was utilized for the quantitative assessment of the environmental change characteristics of the railway area. Based on the results of BO-XGBoost model compensation for anemometer measurement errors, the goal is to improve the accurate monitoring and control of wind speed during high-speed rail operation. In addition, error analysis was conducted to assess the impact of wind observations on enhancing the operational safety of high-speed trains in extreme weather conditions. This article provides a scientific basis for ensuring the reliability and stability of the transportation system.



## 2 Methodology

### 2.1 Two-dimensional ultrasonic anemometers

Two-dimensional ultrasonic anemometers typically utilize the Time-of-Flight (TOF) method for wind measurement. TOF

is a ranging technique that determines the distance between the sensor and an object by analyzing the time interval between

the transmission and reception of ultrasonic pulses. This method offers superior accuracy and resolution, enabling the detection

of minute wind speed variations. It also facilitates elevated sampling frequencies for real-time monitoring of wind speed

changes (Stellinga et al., 2021).

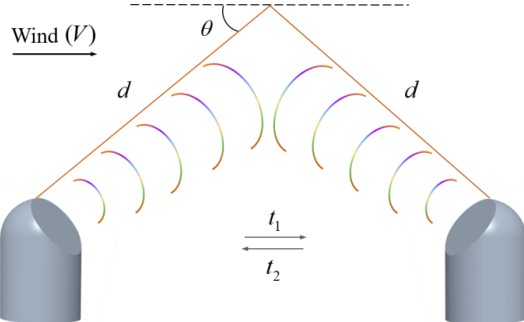

**Figure 1. Diagram of the TOF principle, illustrating the acoustic sensor, ultrasonic pulse channel and wind direction.**

As shown in Figure 1, the time the sensor emits pulses along the wind direction is denoted as $t_1$, and the time it receives

the backscatter pulses is denoted as $t_2$. The relationship between $t_1$ and $t_2$ is expressed as the following equations:

$$t_1 = \frac{d/\cos\theta}{k+V\cos\theta},\qquad\qquad(1)$$

$$t_2 = \frac{d/\cos\theta}{k-V\cos\theta},\qquad\qquad(2)$$

Where $k$ is regarded as the propagation speed of the pulse signal in a windless state. $d$ is the pulse channel length. $\theta$ is

the Angle between the plane and the pulse channel. $V$ is the wind speed. The receiving time difference of the acoustic signal

can be regarded as $\Delta t = |t_1 - t_2|$. Since $k \gg V$, the final wind speed $V$ can be obtained after simplification:

$$V = \frac{\Delta t}{2d\cos\theta} \cdot k^2,\qquad\qquad(3)$$

According to the formula above, the shadow effect is a primary factor contributing to errors in anemometer readings. The

accuracy of the time difference method will decrease if foreign objects obstruct the pulse path.

### 2.2 Computational Fluid Dynamics

CFD serves as a tool for quantifying physical constraints, allowing for an intuitive perception of flow field distribution

characteristics through flow field visualization (Yang et al., 2024a; Yang et al., 2023). In this article, CFD is utilized as a

method for simulating wind fields in railway environments.

An appropriate model is crucial for numerical simulations. Assuming the experiment is performed in a standard state at



a temperature of 25 °C, the flow is considered to be a fluid with a density of 1.1614 kg/m³ and a dynamic viscosity of 1.5898

× 10$^{-35}$ m²/s. Based on laboratory measurements, the inlet flow velocity of air is set to be greater than 13 m/s. According to the

formula for calculating the Reynolds number $Re$:

$$Re = \frac{VL}{\mu}, \tag{4}$$

Where $V$ is the average velocity at the inlet of the airflow. $L$ is the characteristic length. $\mu$ is the dynamic viscosity of the

air. According to the classification of fluid flow, the flow state under extreme high-wind conditions is turbulent. The numerical

solutions for the turbulent kinetic energy $k$ and dissipation rate $\varepsilon$ are solved by the $k - \varepsilon$ model (Equations (5-6)).

$$\frac{\partial \rho U_j k}{\partial x_j} = \frac{\partial}{\partial x_j}[(\mu + \frac{\mu_t}{\sigma_k})\frac{\partial k}{\partial x_j}] + P_k - \rho\varepsilon + P_{kb} , \tag{5}$$

$$\frac{\partial \rho U_j \varepsilon}{\partial x_j} = \frac{\partial}{\partial x_j}[(\mu + \frac{\mu_t}{\sigma_\varepsilon})\frac{\partial \varepsilon}{\partial x_j}] + \frac{\varepsilon}{k}(C_{\varepsilon 1}P_k + C_{\varepsilon 1}C_{\varepsilon 3}P_{\varepsilon b}) - C_{\varepsilon 2} , \tag{6}$$

Where $\mu_t$ represents the turbulent viscosity coefficient of the gas, $P_k$ denotes the turbulent kinetic energy induced by

velocity gradients and $P_{kb}$ stands for the steady-state kinetic energy. In the standard state, the remaining parameters in the air

are usually set in the fluid dynamics software: $\{C_{\varepsilon 1} = 1.44, \ C_{\varepsilon 2} = 1.92, C_{\varepsilon 3} = 0.09, \sigma_k = 1, \sigma_\varepsilon = 1.3\}$.

**2.3 Extreme Gradient Boosting**

Extreme Gradient Boosting (XGBoost) is a powerful machine learning algorithm, particularly excelling in data modeling

and prediction tasks (Ma et al., 2021; Sagi and Rokach, 2021). The specific principles of XGBoost are introduced in Figure 2.

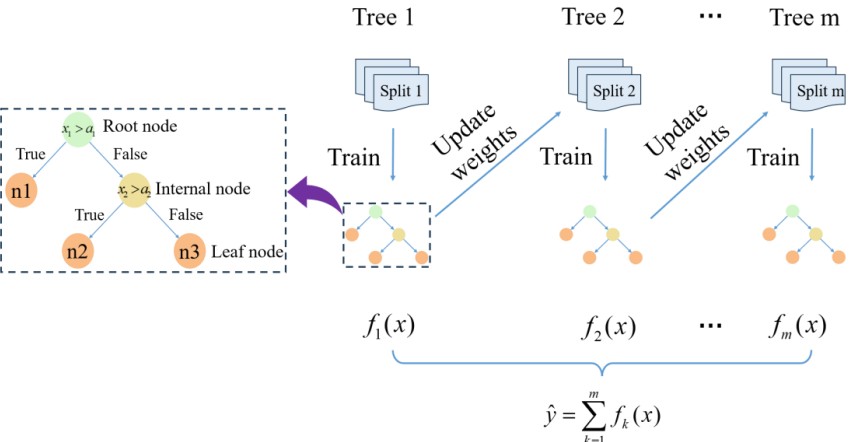

**Figure 2. Diagram illustrating the XGBoost algorithm.**

The estimated output $\hat{y}$ of a gradient-boosting tree model can be represented as the sum of the predicted scores $f_k(x_i)$

of all trees.

$\hat{y}_i = \sum_{k=1}^{m} f_k(x_i), f_k \in \Gamma, \tag{7}$

The XGBoost algorithm employs a learning process utilizing $m$ trees (where $f_k$ denotes the $k^{th}$ tree). Herein, a designates





the space of regression trees, $x_i$ denotes the features of the $i^{\text{th}}$ sample. For each leaf node $j$, a predictive score $f_k(x)$, also referred to as leaf weight, is generated. The leaf weight $\omega_j$ represents the regression value of all samples at leaf node $j$ within the tree. If a tree has $T$ leaf nodes, it can be denoted as $j \in \{1, 2, \dots T\}$.

The objective function plays a crucial role in machine learning problems, and its optimization process continues until the reduction of the objective function reaches a finite state. To approximate the function set used in the model, the following regularized objective function is defined:

$$\phi = \sum_{i=1}^{n} l(y_i, \hat{y}_i) + \gamma T + \frac{1}{2}\lambda \sum_{j=1}^{T} \omega_j^2 \,, \tag{8}$$

Where $\phi$ denotes the value of the loss function, $n$ represents the given data samples, $l(y_i, \hat{y}_i)$ signifies the degree of fit

between the training loss function of the model and the training data. $\gamma T + \frac{1}{2}\lambda \sum \omega_j^2$ denotes the regularization term for the complexity of the tree. Within this context, $\gamma$ refers to the degree of tree splitting, $\lambda$ represents the regularization hyperparameter.

### 2.4 Bayesian Optimization of Hyperparameters

The process of selecting hyperparameters in XGBoost can lead to suboptimal model choices due to various parameter

combinations. Bayesian Optimization of Hyperparameters (BO) efficiently explores complex parameter spaces, adapts over iterations, handles noisy functions and requires fewer evaluations for global optimization (Si et al., 2020; van de Schoot et al., 2021; Xiong et al., 2023). To obtain the optimal parameter combination, this study integrated BO for parameter tuning of the XGBoost model. The hyperparameter optimization problem of the XGBoost model via Bayesian theory can be defined as:

$$\max_{x} f_{\text{XGBoost}}(x), x \in \mathbb{R}^d \,, \tag{9}$$

In Equation (9), $x$ represents the hyperparameters of the XGBoost model. The XGBoost model possesses a complex structure and lacks gradient information. $f_{\text{XGBoost}}(x)$ denotes the objective function used to evaluate the performance of the model. $\mathbb{R}$ represents the hyperparameter space within the XGBoost model. $d$ indicates the dimensionality of the hyperparameters to be optimized in the XGBoost model.

### 2.5 Evaluation metrics

Mean Squared Error (MSE), Root Mean Squared Error (RMSE), Mean Absolute Error (MAE) and Coefficient of Determination (R²) are widely implemented metrics for evaluating the performance of predictive models. These performance measures are utilized for analyzing and evaluating the prediction results of machine learning models in experiments.

$$\text{MSE} = \frac{1}{n}\sum_{i=1}^{n} (y_i - x_i)^2 \,, \tag{10}$$

$$\text{RMSE} = \sqrt{\frac{\sum_{t=1}^{n}(y_i - x_i)^2}{n}} \,, \tag{11}$$





$\quad \mathrm{MAE} = \frac{\sum_{i=1}^{n}|y_i - x_i|}{n},$ (12)

$R^2 = 1 - \frac{\sum(y_i - x_i)^2}{\sum(y_i - \bar{x})^2},$ (13)

In Equations (10-13), $y_i$ represents the $i$ th actual value, $x_i$ represents the $i$ th predicted value, $n$ is the number of data

points and $\bar{x}$ is the mean of $x_i$.

**3 The Proposed Quality Control Method**

This study adopts the PD-BX approach to mitigate errors stemming from the anemometer obstruction by the catenary

pillars. As shown in Figure 3, this method of PD-BX establishes a grid model of the railway anemometer and constructs the

physical constraint relationship between the catenary pillar and the anemometers. In the second step, FLUENT is harnessed to

analyze the influence range of the physical relationship and establish a mapping relationship between simulated data and actual

wind speed. Finally, after segmenting the dataset, high-quality calibration of the high-speed rail anemometer model is achieved

through data-driven hyperparameter optimization, utilizing the BO-XGBoost model. Through the comparison of various

evaluation indicators, the model has been validated to significantly enhance the accuracy of wind observations. This method

provides valuable insights for the design and optimization of similar systems in the future.

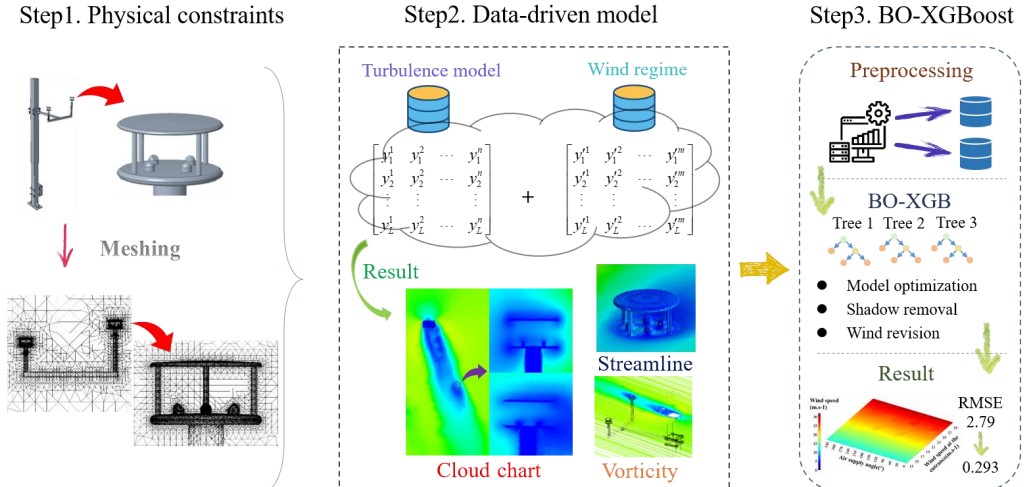

**Figure 3. Flow chart of PD-BX method. This method consists of three steps, including physical constraint establishment, data-driven**
**model implementation, and intelligent algorithm hyperparameter optimization.**

**3.1 Physical modeling**

    Wind speed and direction are typically monitored by an anemometer installed on the bracket attached to the overhead

contact line of the high-speed railway. Figure 4 depicts a structural diagram of a well-established railway anemometer model,

illustrating its intricate design and functional components. The support core of the model is a fixed panel firmly connected to



the overhead catenary struts, providing a stable foundation for the entire system. Protruding from the fixed panel frame is a 1200 mm longitudinal support bar. Reinforcing wings are mounted on the connecting end of the support rod to bolster its resistance from external forces and ensure smooth system operation. In addition, the integration of the transverse support rod completes the interconnected support system of the anemometer model, further enhancing the elasticity and reliability of the anemometer model along the high-speed railway corridor. The vertical rod, perpendicular to the horizontal rod, is installed at both ends of the bracket. The anemometer is installed on the bracket, with the right anemometer positioned above 4 m and the left anemometer below 4 m. The horizontal distance between them is 1000 mm. This configuration allows the anemometer to monitor ambient wind speed without being affected by gusts generated by high-speed trains.

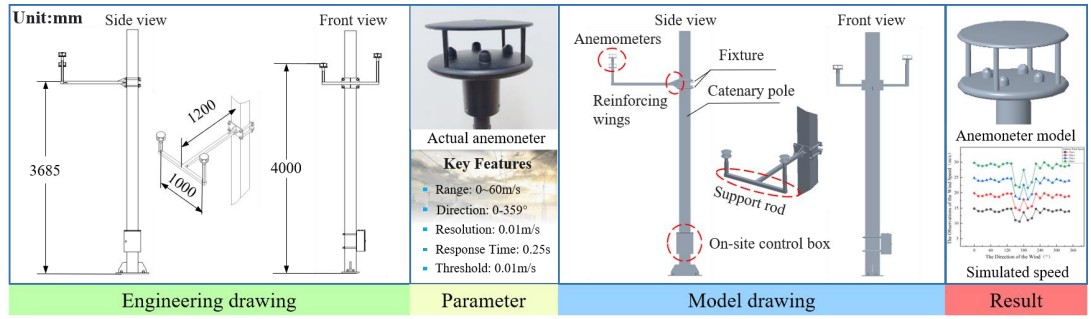

**Figure 4. The railway anemometer, including physical representations, models, parameters and simulated results. All model and physical object dimensions in this figure are based on a 1:1 scale modeling by Creo software.**

## 3.2 Parameter settings of CFD

In the study, the railway anemometer is positioned within a rectangular wind field ten times its size. The windward boundary of the anemometer is distinctly delineated as the entrance boundary, while the leeward boundary is marked as the exit boundary. The remaining four boundaries are designated as slip boundaries, with all boundaries of the anemometer precisely defined as fixed boundaries. Hexahedral non-uniform mesh division is implemented for all surfaces within the model, with most meshes possessing a mass coefficient surpassing 0.5. The inlet boundary is assigned an airflow velocity range of 13-30m/s, while wind direction is confined within 0-360°. All experiments utilize the $k - \varepsilon$ model to simulate the turbulent environment.

### 3.3 Configurations of BO-XGBoost

Through Bayesian optimization theory, the hyperparameters of the XGBoost model have been optimized to mitigate the error of the railway anemometer. Before training begins, the search space for each hyperparameter is defined. Within this space are set the ranges of learning_rate: (0.01, 0.3), max_depth: (3, 10), min_child_weight: (1, 5), n_estimators: (50, 200) and subsample: (0.5, 1.0). Initial hyperparameter values are randomly chosen during the first iteration. Once the iterative algorithm commences, evaluation metrics are inputted. These metrics are combined with historical evolution results and the search space.

Subsequently, the BO-XGBoost algorithm yields the optimization outcomes for each hyperparameter. Then, new
hyperparameter values are received and the next iteration is initiated. This process continues until the error results meet a
predetermined threshold.

## 4 Results and Discussions

### 4.1 Physical constraints visualization and analysis

This experiment employs CFD technology to visualize the constraint relationships of the physical model and conduct an
in-depth analysis of these relationships. Cloud diagrams, vortex diagrams and vector diagrams are utilized as the primary
analytical tools to conduct a comprehensive study of blocking factors, the genesis of formations and the intensity of the impact.
This facilitated a detailed investigation into the specific impact mechanisms affecting the performance of railway anemometers,
thereby furnishing a profound scientific basis for further improving and calibrating wind observations in real-world conditions.

As depicted in Figure 5, the depth and size of the blue shading correspond to the extent of wind speed obstruction. The
blue rectangles indicate the positions where the anemometers are deployed. Below the plan view, detailed illustrations of the
frontal and lateral paths of the anemometers' sonic waves within the dashed frames are presented. The wind resistance of the
anemometer can be intuitively perceived from the cloud images in various wind directions. Concerning the magnitude of the
shadow effect caused by the catenary struts, the shadow effects on the left and right anemometers are most pronounced at wind
directions of 210° and 150°, respectively. Moreover, when contrasting shadow effects caused by different elements, including
sensors and support rods of the anemometer, shadows primarily appear in yellow-green with relatively minor obscuring effects.
In contrast, the shadow of the catenary pillar is predominantly blue, leading to significant obstruction. Due to the prevalent
shadow effects on the anemometer, simulated values generally tend to be lower than the actual flow velocity in the wind field,
highlighting the substantial requirement for error compensation in railway anemometers.

Various wind directions related results are shown in Figure 6. It demonstrates a consistent trend when simulating observed
values of the instrument under different wind speeds, indicating a linear relationship between the observed values of the wind
speed anemometer and the environmental wind speed at the same angle. The dashed box in light pink indicates that the primary
obstruction factor for the wind speed anemometer is the catenary pillar, which is mainly distributed between 180°-225° and
135°-180°, with the most significant obstruction occurring at 165° and 195°. Additionally, the RMSE of the observed values
increases with the rise in wind speed, indicating a growing necessity for error correction of railway anemometers under extreme
wind conditions. Compared with the simulation in this article and the wind tunnel experiment conducted by Assen (Nanjing)
Environment Technology Co.,Ltd. the error rate is less than 0.5%.




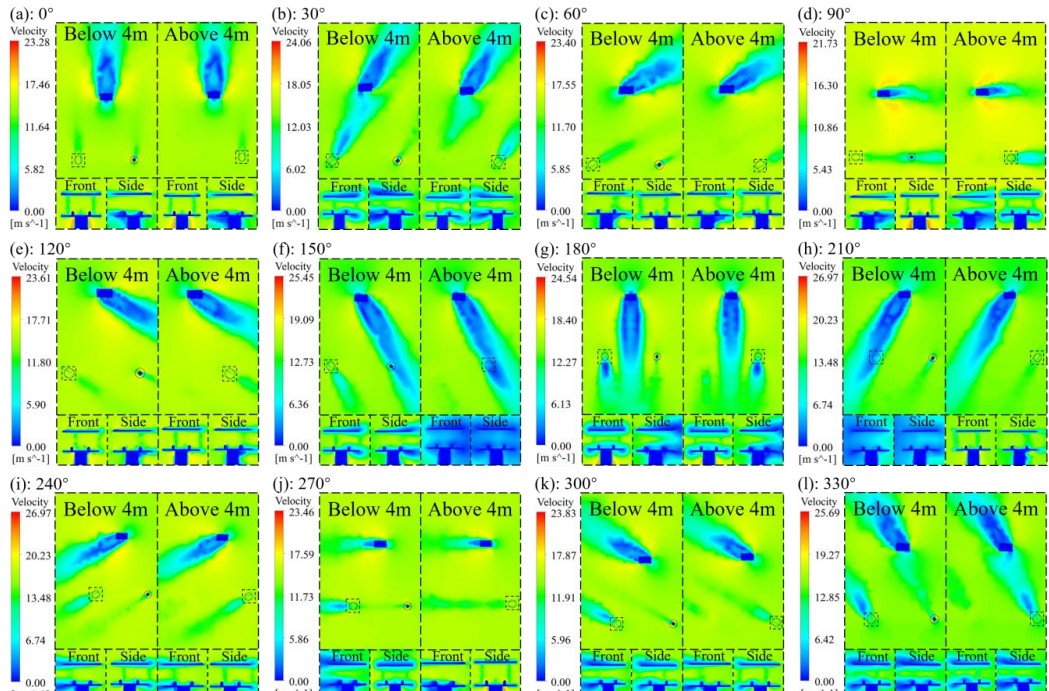

**Figure 5. Cloud maps of the railway anemometer from various angles, with a wind speed threshold set at 15 m/s. The experimental layout includes both the left and right sides, with spatial height divided into maps below and above 4 m, all presented from an overhead perspective angle.**

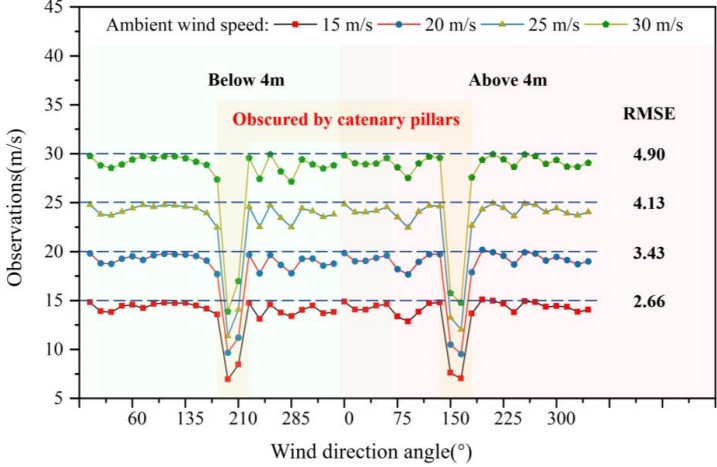

**Figure 6. Wind speed observations. Measurement values of the wind speed anemometer at the main wind speed alarm thresholds are indicated by light green and pink colors, representing observations from anemometers located below and above 4 m, respectively.**

Figure 7 presents wind speed vector maps and vortex maps under extreme shadowing conditions with a wind direction of 165° and a wind speed of 15 m/s. From Figure 7(b-c), it is evident that the anemometer below 4 m is obstructed by its support rods and sensors, with dense vector lines forming a distinct light-shadowed area, resulting in accuracy deviations of the anemometer. In Figure 7(a-c), anemometers positioned above 4 m are significantly obstructed by the catenary pillars, with



vector lines diverging backward on both sides of the center of the pillar's back, forming large vortices. This renders this position 'static' and severely impedes the flow of the wind field, which is the main cause of errors in the wind speed anemometer. Furthermore, the vortex maps in Figure 7(d-e) illustrate the wind vortices around different structural components of the anemometer, with sensor pins obstructed to varying degrees. In Figure 7(d), the sensor pins of the anemometer above 4m appear deep blue, providing further evidence of significant shadow effects at this location.

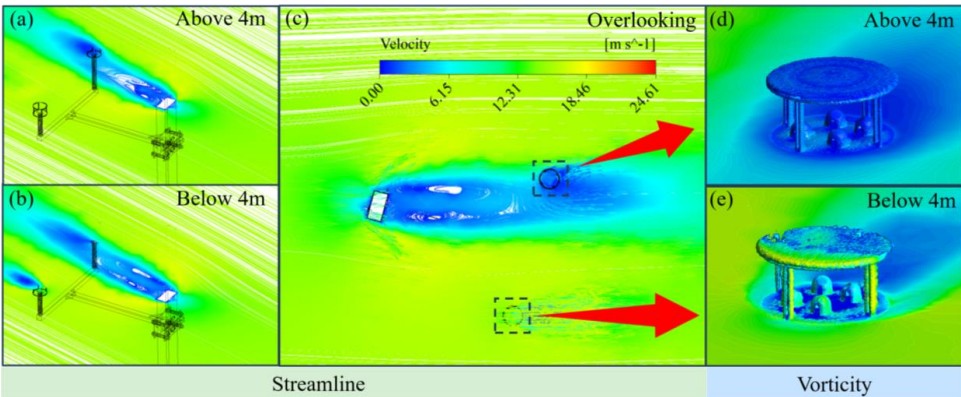


**Figure 7. Multiple wind speed images depicting wind direction at 165° and wind speed of 15 m/s. The results of the fluid dynamics visualization are indicated at the bottom of the figure. (a-c) depicts streamline maps, while (d-e) displays vorticity maps. Additionally, the height is indicated at the top right of the figure.**

### 4.2 Error compensation

To ensure the accuracy of high-speed railway wind measurements, various classical machine learning algorithms were employed to optimize the average measurement results of the anemometer. In this study, wind speed samples were utilized as the dataset, with simulated instrument wind speed and direction serving as input variables and environmental wind speed as the predicted variable. Four-fifths of the total samples were allocated for the training set and one-fifth for the test set. These sets were then inputted into the BO-XGBoost, XGBoost, RF and SVR error correction models to compensate for errors caused

by obstructions of the wind speed and wind direction anemometer. The comparison between the BO-XGBoost model and other models is shown in Figure 8, where BO-XGBoost is distinguished by its smaller numerical errors, reduced outliers and superior evaluation metrics.

In Figure 8(a), the numerical comparison of the model's corrected error against the original data is presented. All four groups of models exhibit significant improvements. The red line segments illustrate the errors post-revision by the BO-

XGBoost model, which are closer to the 0 scale line compared to the other three sets of models. This indicates fewer erratic fluctuations and better numerical outcomes. In Figure 8(b), the BO-XGBoost model demonstrates a narrower error distribution, indicating reduced forecast volatility. The median error closely approximates 0, suggesting a close alignment between the predicted and actual wind speeds. Additionally, upon comparing the error metrics post-correction by different models as outlined in Table 1, it's evident that the BO-XGBoost model tends to yield lower MSE, RMSE and MAE values, while $R^2$



approaches 1, indicative of its superior compensatory effect.

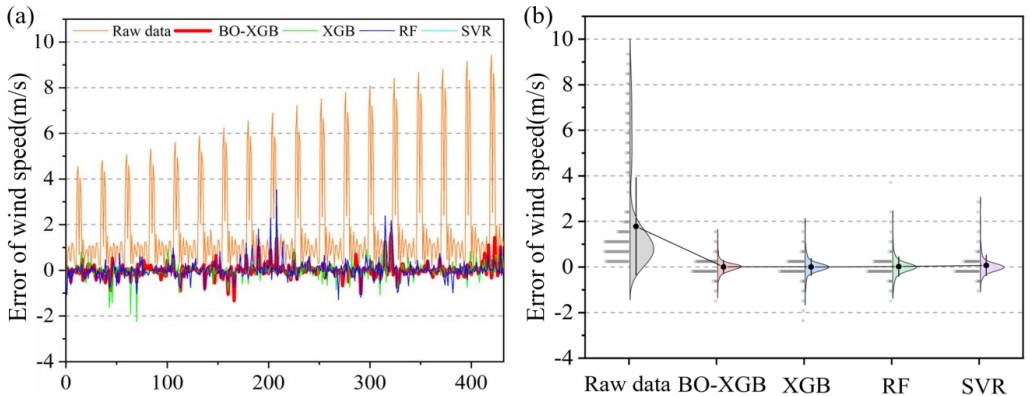

**Figure 8. Comparison of error corrections for different models. (a-b) show the numerical comparison of the errors and the corresponding violin plots.**

**Table 1. Different algorithm correction error indicators**

| Indicators | MSE | RMSE | MAE | $R^2$ |
|---|---|---|---|---|
| BO-XGboost | 0.086 | 0.293 | 0.183 | 0.997 |
| XGboost | 0.180 | 0.424 | 0.242 | 0.993 |
| RF | 0.141 | 0.375 | 0.216 | 0.995 |
| SVR | 0.200 | 0.451 | 0.229 | 0.992 |

The comparison is shown in Figure 9 of BO-XGBoost model results. In Figure 9(a), the elliptical markers cover angular ranges approximately from 140° to 170° and 190° to 220°. Additionally, the rectangular markers predominantly indicate angle intervals around 30°, 75°, 240° and 285°. Furthermore, demonstrating the superiority of the BO-XGBoost model, Figure 9(b) presents a smoother wind speed spectrum, with some spots falling within the reasonable range of railway wind speed error requirements.

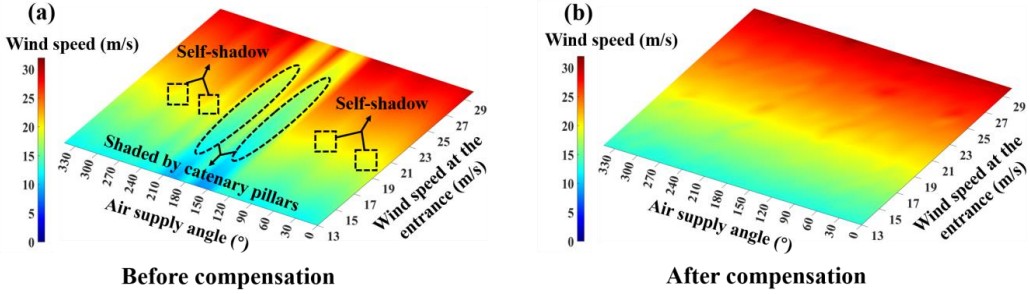

**Figure 9. Comparison of the average observation results of two anemometers before and after compensation by BO-XGB. In (a), the ovals and rectangles depict the angular intervals affected by the catenary pillar and structural elements in the anemometer.**

The red line in Figure 10 corresponds to the particular depiction of wind speed within the dashed line outlined in Figure

9(a). The red curve exhibits significantly lower values compared to the ambient wind speed. Conversely, the compensated blue



curve closely matches the ambient wind speed. This underscores a notable compensatory effect of the BO-XGBoost model in this context. Additionally, owing to the heightened interference from the contact wire pillars observed in Figures 10(a-d), the simulated values are comparatively smaller compared to Figures 10(e-h). The compensation provided by the BO-XGBoost model effectively addresses various obstructive elements, aligning the wind speed with the environmental wind speed and

successfully suppressing abnormal fluctuations. This highlights the superiority of the railway anemometer in handling wind speed data in complex environments.

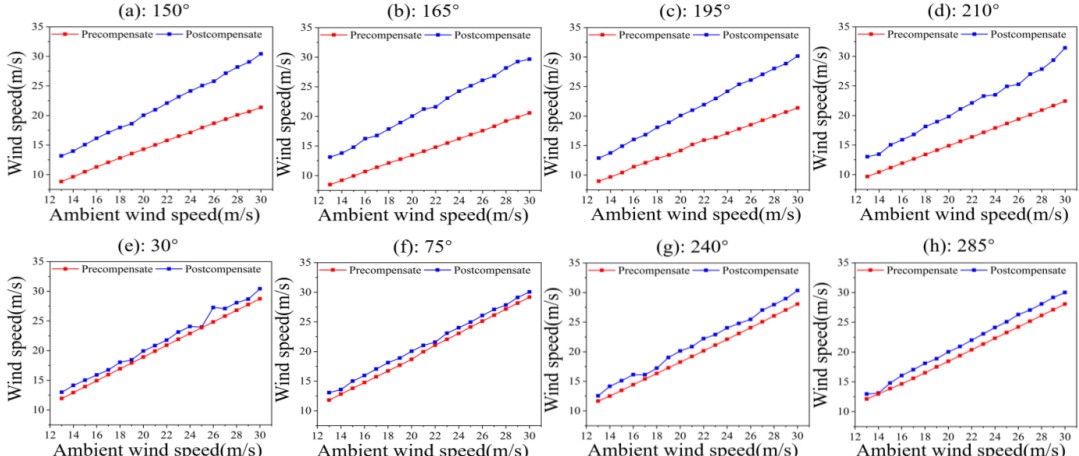

**Figure 10. The specific value of anemometer compensation. The red and blue curves represent the simulated values and the results**
**after compensation via BO-XGBoost.**

### 5 Conclusions

This study utilized the PD-BX method to address the shadow effect induced by catenary supports. The experimental procedure involved quantifying physical constraints based on FLUENT and error compensation employing intelligent models implemented by BO-XGBoost.

During the constraint quantification process, the situation was observed when the anemometer was obstructed from various wind directions. The impact was then subdivided into external obstacles and self-shadowing factors. On the one hand, the research results indicate that the primary cause of anemometer errors is the catenary pillar. In the wind direction intervals of 135°-180° and 180°-225°, the catenary pillar generates a significant shadowing effect on the anemometer. This effect is primarily due to the formation of a large vortex on the leeward side of the catenary support, which obstructs fluid flow. On the

other hand, the shadow cast by the support column and sensor inside the anemometer serves as a secondary obstruction factor.

Besides, the BO-XGBoost showed better compensation results than other models in comparison. It can effectively compensate for anemometer errors induced by shadow factors in certain complex railway environments. The final simulated wind speed RMSE was reduced from 2.79 to less than 0.3, underscoring the outstanding performance of the model in rectifying wind speeds from railway anemometers. Future research will undertake experiments to investigate the obstruction errors of



railway anemometers resulting from other factors in complex high-speed rail environments. This endeavor will enhance our

understanding of the potential impact of high-speed rail systems on wind speed and direction instruments.

*Competing interests.* The authors declare that they have no conflict of interest.

*Data availability.* The original contributions presented in the study are included in the article, further inquiries can be directed

to the corresponding author.

*Acknowledgements.* This research was partially supported by the National Natural Science Foundation of China under Grant

No. 42205150 and 42275156. The Natural Science Foundation of Jiangsu Province, China under Grant No. BK20210661.



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
