# Peer review of "A quality control method based on physical constraints and data-driven collaborative for wind observations along high-speed railway lines"

_EGUsphere, 2024_

## Author Comment (AC1)

**Responses to the comments**

Thanks for your kind comments for our manuscript to Atmospheric Measurement Techniques (ID: https://doi.org/10.5194/egusphere-2024-1006). We appreciate your valuable comments and suggestions to improve it. With regard to your comments and suggestions, we wish to reply as follows:

**Responses to the Reviewer 1**

Reviewer #1: The manuscript presents a PD-BX method to address the shadow effects of catenary pillar and improve the wind speed measurements along the high-speed railway lines. The RMSE of railway anemometer was reduced, providing enhanced accuracy and reliability of wind measurement. The manuscript is well written, and all figures are clear. Please see my comments and suggestions for minor edits below.

General comments:

1. The CFD model was performed under a standard state at 25 °C, and I assume the dry condition. However, under extreme wind conditions, e.g. strong thunderstorms, relative humidity would be high, and heavy rain is also expected. How does authors' model perform under such extreme conditions? Also, in real world, it requires quick response of wind speed under such extreme conditions. How long does it take for authors' model from data processing to wind speed results?

√Response: We sincerely thank the Editors/Reviewers' warm work earnestly and hope that the correction will meet with approval. And We have found that there is little literature on considering other meteorological factors for anemometers in extreme weather, so your opinion is very important to us. As for the results of humidity, the results are shown in Figure 1.

[Figure]

(a) 0%               (b) 20%

(c) 40%               (d) 60%

(d) 80%               (e) 100%

**Figure 1. Cloud images of railway anemometers with different humidity levels, with a wind speed threshold set at 10m/s.**

It can be concluded from Figure 1 that the wind field changes with the variation in humidity, with the minimum value in the basin decreasing as humidity increases. According to Table 1, the measurement results of the anemometer decrease with the increase in humidity, with an error of about 0.2%. However, this error is much smaller

than the case when obstructed by the contact network, and even smaller than the internal shadow effect.

**Table 1. Measurement results of the anemometer under different humidity and wind speed conditions in the flow field**

| humidity | 10m/s | 15m/s | 20m/s | 25m/s | 30m/s |
|----------|-------|-------|-------|-------|-------|
| 0% | 9.93 | 14.87 | 19.84 | 24.81 | 29.80 |
| 20% | 9.91 | 14.78 | 19.77 | 24.74 | 29.69 |
| 40% | 9.87 | 14.72 | 19.68 | 24.66 | 29.56 |
| 60% | 9.84 | 14.68 | 19.64 | 24.59 | 29.47 |
| 80% | 9.79 | 14.61 | 19.59 | 24.45 | 29.38 |
| 100% | 9.77 | 14.58 | 19.54 | 24.41 | 29.30 |

Furthermore, wind speed measurements were not affected by temperature and air pressure. The aforementioned experiments will be arranged in Section 4.1, described as simulating the anemometer's measurement conditions in extreme environments to improve the comprehensiveness of the study. The conclusions will not cause significant changes to the main structure of the article. In the dynamic experiments, the response time varies with wind speed and is approximately the distance from the anemometer to the contact grid support divided by the ambient wind speed, while temperature, air pressure, and humidity have little effect on this response time. (see lines 209-214 in the revised paper)

2. Based on the PD-BX method, authors reduced the uncertainty of wind velocities caused by catenary pillar. Beside velocities, anemometer can also detect wind directions. Do catenary pillars lead to uncertainties of wind directions, e.g. shadow effects. If so, can we also use this PD-BX method to correct the wind directions?

√Response: We agree that making wind revisions will help improve the accuracy of railroad anemometers' measurements in windy fields. However, according to the current technical specifications and standards of China's high-speed railroads, railroad wind observation only focuses on wind speed, i.e., if the wind speed reaches the alarm

threshold for more than 10 seconds, the train will be instructed to decelerate or stop moving into the section. At this stage, the demand for wind direction for high-speed railroad traveling safety is relatively small. However, with the expansion of the scale of China's high-speed railroad, the demand for wind direction monitoring of high-speed railroad is gradually increasing. We are committed to advancing high-speed railroad wind correction experiments in real time based on the specific needs of the railroad.

---

## Author Comment (AC2)

**Responses to the comments**

Thanks for your kind comments for our manuscript to Atmospheric Measurement Techniques (ID: https://doi.org/10.5194/egusphere-2024-1006). We appreciate your valuable comments and suggestions to improve it. With regard to your comments and suggestions, we wish to reply as follows:

**Responses to the Reviewer 1**

Reviewer #2: The manuscript presents a PD-BX method aimed at mitigating the shadow effects of catenary pillars and improving wind speed measurements along high-speed railway lines. Overall, the quality of this paper is good. However, certain sections of the English require improvement. Additionally, the following issues need to be addressed and clarified:

General comments:

1.   Strong winds are a significant factor impacting high-speed rail operations. This paper does not discuss the measures that various countries have implemented to respond to mitigate the effects of strong winds; could the authors provide more specific information on this? Additionally, there are few mentions of rail safety incidents caused by strong winds. It is recommended that the authors include more detailed information on the hazards associated with these incidents.

√Response: We sincerely thank the Editors/Reviewers' warm work earnestly and hope that the correction will meet with approval. In fact, a significant threat to the safety of train operations has always been posed by strong winds. Before the introduction of Japan's "Strong Wind Alarm System," equipped with wind speed prediction capabilities in 2006, over 30 incidents of train derailments and overturns caused by strong winds had been recorded. In 1986, a passenger train on Japan's San'in Line was overturned by strong winds, resulting in 6 deaths and 6 injuries. Similarly, in 1981, a train in India was overturned by strong winds, resulting in over 800 casualties. In China, since the opening of the Lanzhou-Xinjiang Railway, more than 30 incidents of train overturns

caused by strong winds have also occurred. In response to these tragic accidents, railway departments in multiple countries have developed various prediction and warning systems. Up to 2 minutes in advance, the German railway company's "Nowcasting" system can predict peak wind speeds; Italy has established a probabilistic model for wind speed and direction based on data from high-speed railway lines and nearby weather stations; within the next 4 minutes, France can provide predictions for wind speed; and up to 10 minutes in advance, Japan's "Strong Wind Alarm System" can issue warnings and forecasts for strong winds. These technologies have effectively enhanced the safety and reliability of railway operations. The above examples will be included in the introduction to confirm the significance of the experiment. (see lines 23-32 in the revised paper)

2. In the computational fluid dynamics (CFD) analysis presented by the authors, only dry air and standard atmospheric pressure conditions are specified. Comment 1 supplemented the simulation experiments with temperature, humidity and pressure. Please request that the authors to provide additional fluid dynamic parameters for equations (5) and (6).

√Response: In the turbulence model, especially in the k-ε model, the coefficients are empirical parameters typically used to describe turbulence characteristics under different flow conditions. Parameters will vary according to different weather conditions. The specific parameters in this article are: $\{1.2 \leq C_{\varepsilon 1} \leq 1.6,\ 1.6 \leq C_{\varepsilon 2} \leq 2,\ 0.08 \leq C_{\varepsilon 3} \leq 0.1,\ 0.9 \leq \sigma_k \leq 1.1,\ 1.2 \leq \sigma_\varepsilon \leq 1.4\}$. The above results will be presented in Equations 5 and 6. (see lines 134-135 in the revised paper)

3. In the experiment, the authors employed machine learning algorithms to reduce errors. Why not consider simple and convenient physical methods to reduce errors? For example, adjusting the relative position between the anemometer and the catenary support structure could be effective.

√Response: Thank you for your feedback. Compared to traditional physical methods, the physically constrained data-driven approach is more suitable for measuring wind

speed in railway situations. This feedback will be analyzed from the following two aspects. On one hand, the Chinese railway authorities stipulate that wind speed instruments should be installed at a height of either above or below 4 meters relative to the ground. This design not only prevents ground airflow from affecting accuracy but also considers the impact of crosswinds generated by passing trains. On the other hand, using physical adjustment methods cannot completely eliminate shadow effects. Therefore, PD-BX method is the best solution for addressing measurement errors in railway anemometers.

4. There is a point of confusion in Section 4 regarding the results and analysis. Generally, the critical limit for wind speed is set at 15 m/s. However, in Figures 10 and 11, it appears that this article utilities 13 m/s as the threshold for wind speed alerts. Could the authors clarify the reasoning behind this discrepancy?

√Response: Thank you for your question. In our experiments, we found that the contact network support structure not only casts a shadow effect on the anemometer, but under certain conditions, we also observed an acceleration effect of the anemometer. For example, when the wind direction was at 195° or 210°, the readings from the single-sided anemometer exceeded the ambient wind speed. As a precaution to prevent the system from triggering false alarms, we determined that setting the ambient wind speed between 13 and 30 m/s is the optimal range for simulating environmental conditions.